# Factors Affecting Consumers' Alternative Meats Buying Intentions: Plant-Based Meat Alternative and Cultured Meat

**Jihee Hwang [1] , Jihye You [1], Junghoon Moon [1,\*] and Jaeseok Jeong [2]**

[1]   Food Biz. Lab, Program in Regional Information, Seoul National University, Gwanak-gu 151-742, Korea; goodday5@snu.ac.kr (J.H.); Jhyou329@farmair.co (J.Y.)

[2]   Graduate School of Pan-Pacific International Studies, Kyung Hee University Global Campus, Seochon-dong 17104, Korea; profjeong@khu.ac.kr

\*   Correspondence: moonj@snu.ac.kr; Tel.: +82-2-880-4722

**Abstract:** Consumers have started to become aware of the negative aspects of conventional meat, including concerns about environmental issues, animal welfare, and consumer health. Alternative meats (i.e., cultured meat and plant-based meat alternatives) have been introduced recently to address these problems, and the rapid growth of the alternative meat market could pose a threat to the conventional meat market. It is necessary to identify the features of alternative meat that affect consumers' purchasing intentions. Thus, we aimed to: (1) explore the positive and negative feelings toward alternative meat and (2) compare the differences in factors influencing alternative meat buying intentions. This study conducted an online survey with Korean participants in two separate sections (cultured meat: n = 513; plant-based meat alternatives: n = 504), and relationships between the variables and willingness to buy were analyzed using the partial least squares method. The results showed that sustainability and food neophobia are two of the different factors, and food curiosity, unnaturalness, and distrust of biotechnology are the common factors affecting consumers' purchasing choice. The results of this study provide useful guidelines for effective promotional messages about cultured meat, plant-based meat alternatives, and conventional meat marketers focusing on the positive and negative aspects of significant factors.

**Keywords:** sustainable food product; alternative meat; cultured meat; plant-based meat alternatives; ambivalence; willingness to buy; acceptance

## 1. Introduction

With the growth in meat consumption, concerns about meat production systems are on the rise as well. Many studies investigating conventional meat production systems have found that they utilize high amounts of energy, land, and water [1,2]. To produce large amounts of farm-grown meat, many animals need to live indoors under strictly controlled conditions and to be slaughtered, creating ethical issues [3,4]. As consumers have learned about these issues, they feel conflicted when eating meat and want to buy sustainable meat [2,3,5]. Many individuals want to eat meat but do not want it to be linked to any moral, health, or animal welfare issues, creating the so-called "meat paradox" [6]. Due to conflicts between behaviors and attitudes, new food technologies have been introduced to meet the need for eco-friendly or animal welfare–conscious foods [7]. Alternative meat is one of the biggest issues arising in food technologies [8], and it is proposed as an opportunity to address some of the problems created by conventional meat production and consumption [9].

Meat consumption is expected to continue to increase with population growth. There are several alternative sources of protein, including plant-based, cell-based, and insect-based products, to meet

the demand for protein [10,11]. In particular, plant-based meat alternatives and cultured (cell-based) meat are in the spotlight as substitute meat. Cultured meat is defined as a meat substitute made biotechnologically in a laboratory [12]. This novel meat, also called "lab grown meat", "in vitro", or "clean meat", is produced through the in vitro culture of animal muscle cells [13]. Since cultured meat production has not been the dominant approach, many institutions are still introducing and testing various aspects of various methods [7,14] Plant-based meat alternatives mean foods containing only plant-based proteins like pea, soy, coconut oil, and so forth [15]. While cultured meat is not commercialized yet [14], meat made with heme-containing protein from the roots of soy plants (the meat of Impossible Foods (Redwood City, CA, USA), which produces plant-based burgers and sausage) and insect-based foods are already on sale. In reality, the plant-based meat alternative industry has already had some success in the burger patty field [16]. The goal of alternative meat is to satisfy meat-eaters, not vegans, in terms of taste, texture, and appearance [17]. Therefore, it might be a threat to the farm-grown meat market as a supplement.

A literature review found that many articles have actively studied the acceptance of alternative meat. Wilks et al. [18] found the psychological factors affecting willingness to buy (WTB) only focusing on cultured meat, and Bryant and Barnett [19] reviewed papers related to willingness to eat it. Slade [14] conducted experiments inquiring about participants' preferences among cultured meat burgers, plant-based meat burgers, and conventional meat burgers. Very few studies, however, have investigated the psychological acceptance of cultured meat and plant-based meat alternatives integrally. Moreover, no paper has tested the psychological factors affecting WTB alternative meat in Korea. Thus we hypothesized that the factors that affect consumers' WTB cultured and plant-based meat alternatives would be different depending on their perception of the products even though they were introduced for the same purpose (i.e., consuming sustainable meat). Culture heavily influences meat consumption style, and this study helps to explain alternative meat acceptance in Korea. This study has two goals. The first is to investigate the ambivalent (positive and negative) attitudes towards alternative meat. The second is to compare the different factors affecting WTB cultured meat and plant-based meat alternatives. Our findings could offer several guidelines for cultured, plant-based, and conventional meat marketers to develop their promotional messages appropriately.

## 2. Conceptual Framework and Research Hypotheses

### 2.1. Consumer Ambivalence

Berndsen and Van der Pligt [20] showed that consumers have contradictory tendencies when eating meat. Meat consumption involves complex emotions with the concept of meat paradox. For example, it has both advantages and disadvantages in terms of nutritional attributes and health aspects [21,22] in that it could provide high-quality protein but could also be associated with the risk of chronic diseases [23]. Given these facts, when consuming meat, consumers could have conflicting emotions.

Kaplan [24] called these attitudes ambivalence, which is defined as a psychological state containing both a favorable and an unfavorable attitude toward a given object. Stated differently, positive and negative attitudes are in conflict when people make decisions [24]. Hodson et al. [25] addressed that people who have higher ambivalence toward the target are open to suggestions because they are in an unstable state. In the case of alternative meat, many studies have found that consumers also have both positive and negative attitudes toward them. That is, consumers' behaviors related to alternative meat could change depending on what information they receive [26].

The main issue pertaining to the acceptance of alternative meat is about perceived personal and societal benefits and risks. On one hand, previous literature indicates some positive aspects of alternative meats. In terms of sustainable agriculture, some researchers found that alternative meats are more sustainable than farm-grown meat products considering the environment, animal welfare, and natural resources [27,28], and they are good for society. Moreover, alternative meat does not require antibiotics or hormones to achieve mass meat production [29]. The new type of meat is an

innovative food, so consumers feel curious about it [30]. On the other hand, prior research has also examined the following negative aspects of alternatives. This novel food applies agro-food technology, so it is regarded as an unnatural product. The perception that such meat is unnatural or man-made will have a negative effect on consumers' acceptance of alternative meat [31]. Although these alternative meats have gained approval for regulation from the Food and Drug Administration (FDA) and the United States Department of Agriculture (USDA) [32], many consumers are still afraid of the potential danger of technology [33]. Furthermore, alternative meat is not familiar to consumers, so consumers can experience food neophobia [34]. Integrating previous studies allows the features of alternative meat to be classified into three overall categories: ethical features, including sustainability (positive) and unnaturalness (negative); food safety issues, including its being drug-free (positive) and distrust of biotechnology (negative); and consumers' initial reactions, including food curiosity (positive) and food neophobia (negative). This study adopts the ambivalence concept in each category (e.g., ethical viewpoint, food safety, initial reaction), including both positive and negative cognitions to demonstrate the construct validity. Figure 1 demonstrates these framework concepts.

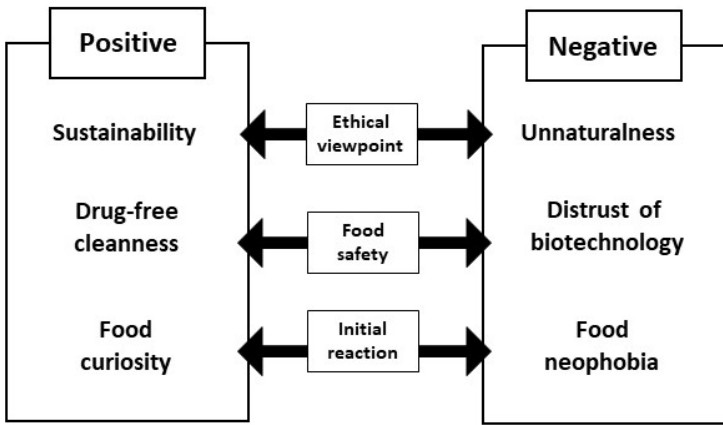

**Figure 1.** Consumers' cognitive conflicts about alternative meat.

## 2.2. Ethical Viewpoint

Livestock production faces some ethical issues, including animal welfare and environmental impacts [35,36]. As a substitute for conventional meat, cultured meat and plant-based meat alternatives have been introduced to solve many problems related to conventional meat consumption. Moreover, ethical consumption is one of the important factors when consumers choose foods, including meat [35,36]. Therefore, we considered ethical viewpoints as important factors in consumer attitudes toward alternative meat. From the perspective of sustainable agriculture, alternative meat may be considered a sustainable product. However, some argue that eating alternative meat can be considered an unnatural practice that removes us from nature [17], which is related to ethical issues, including perceived risks to the intrinsic value of the animals [37]. In this study, the category of ethical viewpoint is divided into the concept of sustainability (positive cognitive) and unnaturalness (negative cognitive).

### 2.2.1. Sustainability

In terms of sustainability, livestock is regarded as a major cause of greenhouse gas emissions today [38]. France has provided consumer advice on recognizing the environmental impacts of meat consumption and choosing sustainable food [39]. The U.S. Department of Agriculture states that 10.378 billion animals have been killed for food products in 2007. Moreover, factory farms are known to create stress and lack adequate sunlight and ventilation, keeping animals in the most wretched conditions [40]. Thus, some scientists argue that current livestock systems do not appear to be sustainable in the long term [41]. In order to solve many ethical problems pertaining to conventional meat, alternative meat has been introduced. [9,42]. For example, Harper and Makatouni [43] found that it does not

deal with animal slaughter for meat production and is related to eco-friendly products, reducing the carbon footprint from livestock. An increasing amount of evidence, including the demand for animal-friendly food, has shown how much consumers are concerned about animal welfare and environmental pollution these days [43,44]. Thus we propose the following hypotheses:

**Hypotheses (H1a).** *Sustainability will positively affect willingness to buy cultured meat.*

**Hypotheses (H1b).** *Sustainability will positively affect willingness to buy plant-based meat alternatives.*

### 2.2.2. Unnaturalness

There are also negative opinions from an ethical viewpoint. As already well-known, consumers prefer natural products and expect them to have far less potential perils than unnatural products [45,46]. In reality, it is well known that alternative meat is very undesirable to consumers with antipathy towards something made in a laboratory using biotechnology. Many consumers regard alternative meat as being unnatural, artificial, synthetic food, and even contrary to nature [33,42,47]. Some believe the way alternative meat is made could have a dangerous impact on people or the environment [48]. Moreover, alternative meat is becoming undistinguishable from conventional meat, so it is possible for people to feel frightened beyond their familiar feelings; this is called the "uncanny valley" concept [49]. So perceived unnaturalness would make many consumers feel alternative meat is inherently unethical [50]. Thus:

**Hypotheses (H2a).** *Unnaturalness will negatively affect willingness to buy cultured meat.*

**Hypotheses (H2b).** *Unnaturalness will negatively affect willingness to buy plant-based meat alternatives.*

### 2.3. Food Safety

Due to technological and environmental changes [51], there are many kinds of food, and consumers can choose whatever they want to eat [52] within strict parameters, including food safety [53]. Piggott and Marsh [54] investigated whether consumers' purchase of beef, pork, and poultry could be affected by food safety concerns. Especially in livestock production, food safety is one of the most important factors in farming methods [55]. Livestock are generally grown with synthetic pesticides, antibiotics, and growth hormones [56]. The usage of antibiotics could cause many problems, including the transmission and spread of disease epidemics and swine influenza, which are serious threats to global health [57]. Becoming more aware of how conventional meat is made, consumers desire to purchase meat without concerns about the many kinds of drugs injected into animals and the side effects of conventional farming practices [58–60]. In general, chemicals, such as pesticides and hormones, and food technology are closely related to food safety issues, which could directly affect individuals' health. Thus, in this study, we classify food safety issues as drug-free cleanness (positive cognitive) and distrust of biotechnology (negative cognitive).

### 2.3.1. Drug-Free Cleanness

Chemical products that remain as residues in foods of animal origin include drug, pesticides, and so on [61]. Livestock is generally grown with synthetic pesticides, antibiotics, and growth hormones. Alternative meat does not require the antibiotics and growth hormones used in raising animals for slaughter [19], so cultured meat is also called clean meat [8,62]. In addition, plant-based meat alternatives are made from plant ingredients [15]. Consumers are concerned about the drug injections given to animals because of individuals' health [63].Thus, we formulated the following hypotheses:

**Hypotheses (H3a).** *Drug-free cleanness will positively affect willingness to buy cultured meat.*

**Hypotheses (H3b).** *Drug-free cleanness will positively affect willingness to buy plant-based meat alternatives.*

### 2.3.2. Distrust of Biotechnology

Although alternative meat has drug-free cleanness as a sufficient strength, consumers' concerns about food safety are increasing based on technological and environmental changes [51]. Science and technology are necessary not only to raise living standards but also to develop a sustainable world [64]. However, there are some issues with new technologies that might have potential risks and side effects for human health and the environment [64]. Especially concerning the technology applied to food, consumers are pickier about what they eat [65,66] because their distrust of biotechnology makes them worried about potential long-term effects of technology [67]. Cultured meat is made from a laboratory [28,33,68]. Further, plant-based meat ferments yeast and is genetically engineered to extract a heme protein from plants. Therefore, according to previous studies, consumers might feel uncomfortable about purchasing alternative meat. Thus:

**Hypotheses (H4a).** *Distrust of biotechnology will negatively affect willingness to buy cultured meat.*

**Hypotheses (H4b).** *Distrust of biotechnology will negatively affect willingness to buy plant-based meat alternatives.*

### 2.4. Initial Reaction

The first reactions are important factors when consumers are confronted with novel foods [69]. There seem to be two attitudes toward new food: food curiosity and food neophobia [70]. Likewise, there are some findings that initial reactions, including curiosity and neophobia, could affect WTB alternative meat. Since the alternative meat market is new, consumers' first impressions are important. Therefore, we use food curiosity and neophobia as criteria of the initial reaction.

### 2.4.1. Food Curiosity

Food curiosity is defined as "the ability of the eater to want to know everything that is related to food, whether at the stage of production, processing and consumption" [71] (p. 7). Van der Weele and Driessen [72] have shown how consumers express their first responses to cultured meat. Their results indicated that 'Wow!' comprised 40–80% of initial responses, and 'Interesting, but' comprised 10–35%. Berlyne [73] suggested that consumers are in a conflicted state as they are easily affected by curiosity concerning individual characteristics and novelty for its capacity to stimulate. This study found that consumers are ambivalent regarding alternative meat, implying instability, so curiosity may be one of the factors affecting consumers' WTB. Thus:

**Hypotheses (H5a).** *Food curiosity will positively affect willingness to buy cultured meat.*

**Hypotheses (H5b).** *Food curiosity will positively affect willingness to buy plant-based meat alternatives.*

### 2.4.2. Food Neophobia

Concerning food neophobia, Pliner and Hobden defined food neophobia as "a reluctance to eat and/or avoidance of novel foods" ([74], p. 105), which is related to the acceptance of new foods. Food neophobia is regarded as an attitude formed by consumers to protect themselves from uncertain food [70], and it could be a barrier to an initial trial. According to Hoek et al. [75], people who are non-consumers of alternative meat have higher degrees of food neophobia than light, medium, and heavy consumers. Especially in terms of judging a new biotechnological process to be morally and legally permissible, people pay more attention to unappealing factors [76]. Cultured meat and plant-based meat alternatives are a new category in the food market, so the idea of food neophobia could negatively influence consumers' buying intentions as a defense system. Thus:

**Hypotheses (H6a).** *Food neophobia will negatively affect willingness to buy cultured meat.*

**Hypotheses (H6b).** *Food neophobia will negatively affect willingness to buy plant-based meat alternatives.*

Therefore, our research model is as follows (Figure 2).

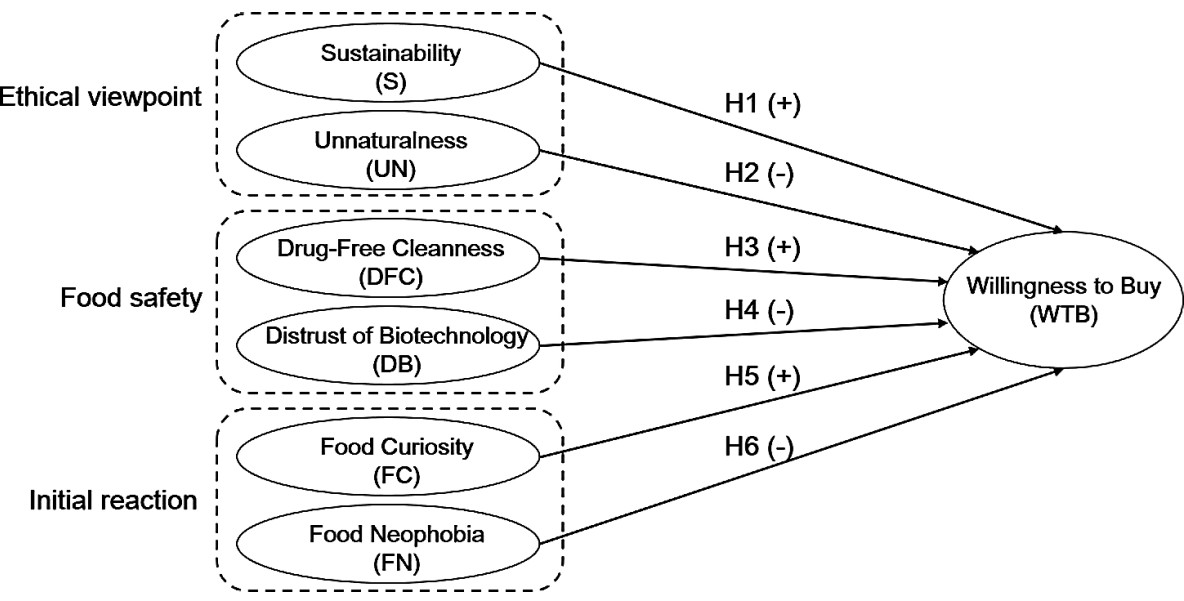

**Figure 2.** Research model.

## 3. Methodology

Data were collected for four days in August 2018 with a support of EMBRAIN, an online survey institution, and a survey link was posted on the online bulletin board. After selecting responses from participants only aged 20 s to 60 s, 1017 responses were used (eight participants dropped out). Respondents' prior consent was conducted before the online survey, and all members of panel agreed to participate. Following Tsang et al. [77]'s guidelines, the questionnaire was translated to Korean by the authors and verified by a balanced bilingual English-Korean speaker. Before conducting the main survey, we constituted an offline expert committee with the master and doctoral students in order to review the questionnaire and asked them to check the inaccurate items. After changing the words in the questionnaire based on the comments, a preliminary pilot testing with 11 respondents to clarify the meaning of the sentences. Then, in order to improve the reliability and determine the way to move a large-scale, we administered the pilot survey twice, once with 30 respondents and once with 40 respondents, on August 1, 2018, only for the case of cultured meat. Then pilot test analysis was conducted with 56 participants, excluding 24 who did not complete the survey.

A questionnaire was developed to assess general attitudes toward livestock products and foods. The survey consisted of two separate questionnaires for cultured meat (n = 513) and plant-based meat alternative (n = 504). The participants completed the questionnaires by themselves, and the survey took about 10 min per respondent. Basic information about alternative meat was provided before the survey. All questions were answered using a seven-point Likert scale (1 = "strongly disagree" to 7 = "strongly agree").

The questionnaire had three parts. Part one dealt with common features regarded as important factors when participants purchase meat. The items were based on a literature review describing the features of farm-grown and alternative meat. The sustainability of animal welfare scale, dealing with attitudes toward factory farming, was based on Anomaly [40]. The unnaturalness scale based on Welin and Van der Weele [17] described attitudes toward the way alternative meat is made. Drug-free cleanness was measured by three items representing consumers' concerns about drugs injected into conventional meat [78]. Further, distrust of biotechnology was measured by two items representing consumers' perception of technology [65]. The food curiosity scale was adapted from UEDA [71], and the food neophobia scale was selected from Pliner and Hobden [74]. Appendix A shows the details. The survey items include reverse coding questions. For the case of the reverse coding, all items were

converted to opposite directions by subtracting the corresponding response value from 8 points on a seven-point Likert scale. After matching the direction of survey items, an analysis was conducted. Part two consisted of WTB, focusing on alternative meat itself, and included a measurement of political orientation, which acted as the "market variable". We used this market variable to assess bias. In part three, respondents were asked to indicate their age, sex, job, educational level, income, sum of red meat intake, and knowledge of alternative meat.

## 4. Results

### 4.1. Demographics

The participants ranged in age from 20 s to 60 s years. The samples were composed of 53% to 47% and 51% to 49% male to female participants, respectively, for the cultured meat and alternative surveys. All participants were over the age of 20 years with a range of education levels. Table 1 shows the demographics results.

**Table 1.** Demographic composition of the participants.

| Variables | Items | Cultured Meat | | Plant-Based Meat Alternative | |
| --- | --- | --- | --- | --- | --- |
| | | Number | Percentage | Number | Percentage |
| Sex | Male | 274 | 53.4% | 256 | 50.8% |
| | Female | 239 | 46.6% | 248 | 49.2% |
| Age | 20–29 years old | 109 | 21.2% | 110 | 21.8% |
| | 30–39 years old | 111 | 21.6% | 115 | 22.8% |
| | 40–49 years old | 119 | 23.2% | 109 | 21.6% |
| | 50–59 years old | 101 | 19.7% | 96 | 19.0% |
| | 60–69 years old | 73 | 14.2% | 74 | 14.7% |
| Education level | High school diploma or less | 72 | 14.0% | 66 | 13.1% |
| | Undergraduate | 38 | 7.4% | 38 | 7.5% |
| | College graduate | 310 | 60.4% | 307 | 60.9% |
| | Graduate student or more | 93 | 18.2% | 93 | 18.5% |

### 4.2. Measurement Assessment

To assess the validity of each individual item, this study examined internal consistency measures using the composite reliability. The composite reliability of the measures ranged from 0.749 to 0.959, which are higher than 0.7, indicating the construct had reasonable internal consistency [79]. To assess the construct validity, convergent validity and discriminant validity were used. For convergent validity, composite reliability (CR), factor loadings, and average variance extracted (AVE) were assessed. Table 2 shows individual item factor loadings were greater than 0.6 [80], and the AVE of each construct was similar to or greater than 0.5 [81]. To ensure the discriminant validity, the square root of the AVE should be greater than its correlations with the others [82]. Table 3 shows all demonstrated sufficient discriminant validity.

**Table 2.** Loadings of survey items and description of variables.

| Latent Variables | Items | Cultured Meat | | | | Plant-Based Meat Alternative | | | |
| --- | --- | --- | --- | --- | --- | --- | --- | --- | --- |
| | | Factor Loadings | t-Statistic | Composite Reliability | AVE | Factor Loadings | t-Statistic | Composite Reliability | AVE |
| Sustainability (S) | S_1 | 0.828 | 7.8811 | | | 0.822 | 24.4907 | | |
| | S_2 | 0.835 | 9.1121 | 0.864 | 0.679 | 0.863 | 24.3023 | 0.876 | 0.703 |
| | S_3 | 0.808 | 6.694 | | | 0.829 | 22.4426 | | |
| Unnaturalness (UN) | UN_1 | 0.891 | 74.8589 | | | 0.709 | 12.6567 | | |
| | UN_2 | 0.839 | 37.4918 | 0.829 | 0.624 | 0.702 | 11.6095 | 0.749 | 0.499 |
| | UN_3 | 0.612 | 11.6147 | | | 0.707 | 11.7316 | | |

**Table 2.** *Cont.*

| Latent Variables | Items | Cultured Meat | | | | Plant-Based Meat Alternative | | | |
|---|---|---|---|---|---|---|---|---|---|
| | | Factor Loadings | t-Statistic | Composite Reliability | AVE | Factor Loadings | t-Statistic | Composite Reliability | AVE |
| Drug-free cleanness (DFC) | DFC_1 | 0.735 | 3.1854 | 0.896 | 0.744 | 0.923 | 59.7456 | 0.955 | 0.876 |
| | DFC_2 | 0.991 | 3.2479 | | | 0.939 | 106.8481 | | |
| | DFC_3 | 0.843 | 3.6683 | | | 0.947 | 104.5321 | | |
| Distrust of biotechnology (DB) | DB_1 | 0.916 | 59.9473 | 0.883 | 0.790 | 0.842 | 17.4336 | 0.855 | 0.746 |
| | DB_2 | 0.861 | 27.3038 | | | 0.886 | 26.9034 | | |
| Food Curiosity (FC) | FC_1 | 0.883 | 30.5399 | 0.850 | 0.656 | 0.726 | 10.6182 | 0.828 | 0.618 |
| | FC_2 | 0.764 | 14.7327 | | | 0.885 | 26.4494 | | |
| | FC_3 | 0.777 | 16.5003 | | | 0.737 | 11.7641 | | |
| Food neophobia (FN) | FN_1 | 0.832 | 7.2113 | 0.887 | 0.724 | 0.726 | 3.0219 | 0.859 | 0.674 |
| | FN_2 | 0.780 | 8.1644 | | | 0.735 | 3.1164 | | |
| | FN_3 | 0.933 | 11.2198 | | | 0.977 | 3.4951 | | |
| Willingness to Buy (WTB) | WTB_1 | 0.960 | 184.1365 | 0.959 | 0.921 | 0.951 | 124.6362 | 0.943 | 0.893 |
| | WTB_2 | 0.959 | 161.9829 | | | 0.939 | 71.1522 | | |

S: Sustainability; UN: Unnaturalness; DFC: Drug free cleanness; DB: Distrust of biotechnology; FC: Food curiosity; FN: Food neophobia; WTB: Willingness to Buy.

**Table 3.** Correlations among constructs.

| | <Cultured Meat> | | | | | | | <Plant-Based Meat Alternative> | | | | | |
|---|---|---|---|---|---|---|---|---|---|---|---|---|---|
| | S | UN | DFC | DB | FC | FN | | S | UN | DFC | DB | FC | FN |
| S | 0.824 | | | | | | S | 0.838 | | | | | |
| UN | −0.21 | 0.790 | | | | | UN | −0.043 | 0.706 | | | | |
| DFC | 0.474 | −0.177 | 0.863 | | | | DFC | 0.563 | −0.022 | 0.936 | | | |
| DB | −0.215 | 0.409 | −0.079 | 0.889 | | | DB | −0.049 | 0.318 | −0.037 | 0.864 | | |
| FC | 0.139 | −0.209 | 0.11 | −0.249 | 0.810 | | FC | 0.187 | −0.081 | 0.326 | −0.16 | 0.786 | |
| FN | 0.143 | 0.016 | 0.209 | −0.026 | −0.025 | 0.960 | FN | 0.254 | 0.164 | 0.28 | 0.063 | −0.008 | 0.821 |

S: Sustainability; UN: Unnaturalness; DFC: Drug free cleanness; DB: Distrust of biotechnology; FC: Food curiosity; FN: Food neophobia; *Note*. The diagonal elements are the squared roots of the AVEs.

### 4.3. Assessment of the Structural Model

To test the structural model's results, this study conducted the PLS graph. Relationships between the variables were analyzed using the partial least square (PLS) method. PLS analysis can confirm a theory and suggest the key to finding the relationship between the dependent variables and the independent variable [83]. Using PLS, the researcher could bind several measures into one latent variable [82]. The goal of PLS regression was to predict WTB from six characteristics of alternative meat [84]. In this study, we conducted two PLS-models for each alternative meat (cultured meat and plant-based meat alternative). The first model includes only positive aspects (sustainability, drug-free cleanness, and food curiosity) and the second model includes only negative aspects (unnaturalness, distrust of biotechnology, and food neophobia) to eliminate the multicollinearity problems.

The results listed in Table 4 indicate the key factors that affect consumers' WTB. For cultured meat, curiosity in general attitudes toward food was only a strong factor in positive aspects ($R^2 = 0.17$, $p < 0.001$). On the other hand, in terms of negative aspects, the feelings of unnaturalness concerning cultured meat, distrust of biotechnology, and food neophobia were all statistically significant ($R^2 = 0.54$, $p < 0.001$). For plant-based meat alternative, the features of sustainable livestock production ($p < 0.05$) and the degree of food curiosity ($R^2 = 0.11$, $p < 0.001$) were significant. In negative aspects, the feelings of unnaturalness concerning plant-based meat alternative and the distrust of biotechnology were significant ($R^2 = 0.24$, $p < 0.001$).

**Table 4.** Results of linear regression.

| | Cultured Meat | | Plant-Based Meat Alternative | |
|---|---|---|---|---|
| | Path Coefficients | *p*-Value | Path Coefficients | *p*-Value |
| Positive aspects | | | | |
| S- > WTB | 0.096 | 0.100 | 0.128 * | 0.025 |
| DFC- > WTB | 0.074 | 0.447 | 0.09 | 0.148 |
| FC- > WTB | 0.217 *** | 0.000 | 0.191 *** | 0.000 |
| Age- > WTB | −0.028 | 0.499 | −0.045 | 0.361 |
| Sex- > WTB | −0.203 *** | 0.000 | −0.062 | 0.157 |
| Education level- > WTB | −0.047 | 0.261 | −0.087 * | 0.025 |
| Meat intake- > WTB | 0.115 * | 0.014 | 0.044 | 0.496 |
| Knowledge- > WTB | −0.128 ** | 0.003 | −0.094 * | 0.030 |
| R^2 | 0.17 | | 0.11 | |
| Negative aspects | | | | |
| UN- > WTB | −0.557 *** | 0.000 | −0.371 *** | 0.000 |
| DB- > WTB | −0.222 *** | 0.000 | −0.191 *** | 0.000 |
| FN- > WTB | −0.121 *** | 0.000 | −0.032 | 0.596 |
| Age- > WTB | −0.034 | 0.246 | 0.060 | 0.091 |
| Sex- > WTB | −0.122 *** | 0.000 | −0.033 | 0.409 |
| Education level- > WTB | −0.013 | 0.699 | −0.068 | 0.135 |
| Meat intake- > WTB | 0.032 | 0.385 | −0.080 | 0.094 |
| Knowledge- > WTB | −0.072 * | 0.018 | −0.086 * | 0.023 |
| R^2 | 0.54 | | 0.24 | |

S: Sustainability; UN: Unnaturalness; DFC: Drug free cleanness; DB: Distrust of biotechnology; FC: Food curiosity; FN: Food neophobia; WTB: Willingness to buy; *Note*. Control variables: Age, sex, education level, frequency of meat intake, and knowledge of alternative meat. *** $p < 0.001$, ** $p < 0.01$, * $p < 0.05$.

Considering control variables, in this study, alternative meat observed higher WTB in terms of age, sex, education level, frequency of meat intake, and knowledge of alternative meat. The results regarding cultured meat showed that sex and knowledge of it were significant. To be specific, males had higher WTB than females ($p < 0.001$). Further, the participants who knew about it before the survey had higher WTB cultured meat than the participants who did not ($p < 0.01$).

In the case of plant-based meat alternative, educational level and knowledge of plant-based meat alternative were statistically significant. The lower the participants' education level, the more likely they were to buy plant-based meat alternative ($p < 0.05$). Like cultured meat, prior knowledge of plant-based meat alternative had positive effects on WTB ($p < 0.05$).

## 5. Discussion

The results of this study have revealed the main factors influencing alternative meat purchase intention. In particular, we investigated how the factors affecting WTB differed between cultured meat and plant-based meat alternatives. Although many authors have figured out the common objections to and acceptance of each of them, we found few studies focusing on the two kinds of alternative meat integrally. In addition, although some papers contain both kinds of alternative meat [29,34,85] in the case of various countries, this is the first paper that deals with Korean pertaining to consumer psychology. Since meat is one of the most important ingredients affected by culture, we expected that alternative meat acceptances have different results in various countries.

First of all, the major difference between the factors affecting WTB the two alternative types of meat can be observed in the variables of sustainability and food neophobia. Sustainability was not significant in cultured meat, while it was significant in plant-based meat alternatives. That means many people care about sustainable livestock production, but it does not mean that their WTB cultured meat is high. Verbeke et al. [42] indicates that consumers feel few personal benefits such as taste and price but perceive more global benefits. That is, cultured meat is not fit for everyday

consumption practices although people know cultured meat is good for the environment and animal welfare. While many previous studies found cultured meat to be a sustainable source of protein [86,87], we revealed that awareness of its sustainability does not lead to the purchasing of cultured meat. Thus, this study offers a broad understanding of consumer behavior in that it found that the public interest in sustainable agriculture did not affect the intention to purchase cultured meat. On the other hand, the results of the plant-based meat alternatives survey show that people who care about sustainable farming, like non-factory farming, are more likely to buy plant-based meat alternatives. Our results echo Pimentel and Pimentel [1]. That is because it is well known that plant-based diets are more sustainable than meat-based diets [1], so the feature of sustainability would be well-linked to plant-based meat alternatives. Previous research on plant-based products are often explained in relation to sustainability [88,89], and we checked again.

Food neophobia was significant in cultured meat but not in plant-based meat alternatives. For cultured meat, the technology applied to it was novel to the participants; therefore, the results support Wilks et al.'s [18] investigation that distrust in science and food neophobia are all linked to negative cognitions of cultured meat. Indeed, our participants engaged with food neophobia as a negative aspect. In other words, the more consumers have a high score in food neophobia, the less they perceived the benefits of cultured meat. According to Verbeke's [42] results, the consumers' first impressions of cultured meat relate to disgust or perceived unnaturalness. Moreover, it has been found that in not only Korea but also many other countries, food neophobia has a negative effect on the acceptance of cultured meat because cultured meat has not been commercialized in the market, and it seems to be an unfamiliar food. In line with previous research about food neophobia, this study found that consumers still lack knowledge and certainty of technology about cultured meat. On the other hand, for plant-based meat alternatives, food neophobia had no influence on purchase intention. In the Dominican Republic, food neophobia has a negative effect on the purchase of plant-based meat [34], but our results investigated food neophobia has no effect on plant-based meat. The reason for this is that an increasing number of people have become vegetarian and try to eat a plant-based diet or reduce their meat consumption [90]. In line with this trend, it has been a long time since plant-based, meat-like soy meat for vegans was released. For this reason, plant-based meat alternatives could be a familiar food to participants; thus, the variable of food neophobia would not be significant. Therefore, this study helps to explain the relationship between food neophobia and plant-based meat alternatives.

Secondly, we also found that the factors commonly affecting the purchase of both alternative types of meat were food curiosity, unnaturalness, and distrust of biotechnology. Regarding food curiosity, it was significant in both cases. That is, people with high food curiosity are more likely to buy alternative meats. Curiosity is regarded as one of the strongest motivating factors [91]. Therefore, we think that while curiosity might make people try new food, it might be hard to create lasting, long-term buyer–supplier relationships. According to Lang [92], regular meat-eaters do not consider reducing meat consumption but consider adopting new ideas and technologies for healthier eating. Thus, consumers could consume alternative meats regarded as innovative products for their curiosity. Unnaturalness was also a statistically significant factor for the WTB alternative meat. This fact corresponds with Laestadius [50], who concluded that perceived unnaturalness makes consumers feel that alternative meat has an ethical problem. People who think that the way cultured meat and plant-based meat alternatives are made is unethical are reluctant to try alternative meat. Moreover, according to Verbeke et al. [42], unnaturalness was also linked to food safety concerns. In actuality, our study participants who worried about biotechnology displayed low WTB alternative meat. The application of this technology is novel, so more regulations may be needed to dispel distrust.

In terms of control variables, the results showed that the knowledge of alternative meat has a positive effect on each cultured meat and plant-based meat. It means that creating environments where consumers could get related information could be critical because cultured meat has not been commercialized yet. According to Verbeke et al. [33], if consumers received information about alternative meat, then they were more likely to try or purchase cultured meat. Our results support

previous research. Moreover, this study found that males prefer cultured meat than females, and we concluded that males are more adaptable of gene technology based on Siegrist [65]'s results. Furthermore, according to preliminary studies, the higher the frequency of meat consumption, and the higher the meat attachment, the less the number of consumers changes in purchasing behavior with substitute meat [93]. In this study, however, the frequency of meat intake did not necessarily affect the intention to purchase cultured meat and plant-based meat alternative.

Academically, the current study contributes to the relationship between the features of alternative meat and WTB, especially cultured meat and plant-based meat alternatives, which had been investigated separately in the previous body of literature. Moreover, we applied ambivalence theory and separated the features of alternative meat into three categories: ethical viewpoint, food safety, and initial reaction. To our knowledge, no paper has dealt with the factors that affect WTB alternative meat in terms of consumers' psychological ambivalence in Korea. We figured out positive and negative cognitions for each category, focusing on psychology. The view of ambivalence focuses more on complex emotions and enriches the theory of consumers' purchasing psychology. We also found that consumers feel differently between the cultured meat and plant-based meat alternatives that have been introduced for the sustainable consumption of meat. In this regard, the results of this study contribute to consumers' overall perceptions of alternative meats. In addition to that, this study contributes to the understanding of sustainable foods by revealing the different factors that affect consumers when an innovative product is released compared with a product that was previously consumed.

This study has several suggestions for practitioners. We identified how the WTB alternative meat is different depending on what characteristics people care about. The results of this study are important not only to alternative meat marketers but also to conventional meat producers. The growing demand for alternative meat could hurt the conventional meat market. Therefore, we suggest the following practical guidelines from a personalized perspective.

First, the cultured meat marketer: The cultured meat marketer should recognize that consumers who value sustainability are less likely to choose cultured meat; therefore, emphasizing this feature is a less efficient way to promote it. Additionally, consumers significantly respond to food curiosity. Thus, the key selling point is that cultured meat is an interesting food compared with conventional meat. On the other hand, negative aspects, such as unnaturalness, distrust of biotechnology, and food neophobia, are significant factors that decrease consumers' WTB. Therefore, it is important for cultured meat marketers not to mention negative phrases, such as, "it is made in a laboratory or by biotechnology."

Second, the plant-based meat alternative marketer: Consumers respond to ethical viewpoints, including sustainability and unnaturalness. It could be effective to make the customers aware of the "sustainability" and "specialness" of plant-based meat alternatives to commercialize plant-based meat alternatives. For example, words such as "sustainable livestock" and "eco-friendly meat" could emphasize positive aspects for consumers.

Third, the conventional meat marketer: The conventional meat market should be maintained in competition with the alternative meat market by emphasizing the features of alternative meat to which consumers react negatively and by highlighting the advantages of conventional meat. For example, the participants' WTB decreased in this study due to the unnaturalness of alternative meat, so the naturalness of conventional meat should be emphasized to promote current meat production. Moreover, because consumers have a distrust of science and technology, it is good to advertise that conventional meat is natural and does not involve science and technology. Therefore, conventional meat marketers should mention the strong points of livestock meat production in terms of naturalness in flavor and texture compared to the artificial aspects of alternative meat.

Although the results of this study integrate insights and show practical implications, there are several limitations. First, sample bias is a possible error of this study. In Korea, both cultured meat and plant-based meat alternatives are not commercialized yet; thus, a majority of participants did not know much about the advantages and disadvantages of consuming alternative meat. This bias could impair the findings, so it is hard to generalize the results from this study to the whole world. Therefore, future

research should extend this study by conducting a survey and sensory testing in various countries. Second, for the questionnaire we used, some were adapted from existing questionnaires and some were based on previous researches. As the measurement we made based on the literature review didn't verify through prior research, we think it could be a limitation. Therefore, we confirmed the individual items' discrimination validity and convergent validity before the analysis. Third, it is a possible extension of this study to conduct a field observation study. While we conducted an online survey, a more specific study is needed, including the observation of actual alternative meat purchasing behavior in the real market. Moreover, it is hoped that future studies will assess sensory testing and repurchase intention. Fourth, it would be good to extend the concept of alternative meat, including insect-based products, in terms of sustainable meat consumption.

## 6. Conclusions

This study demonstrated that the consumers' buying intentions concerning cultured meat and plant-based meat alternative are different based on concepts of ambivalence. We identified the positive and negative cognitions depending on their perception of cultured meat and plant-based meat alternative attributes. This is the first time the relationship between consumers' purchase behavior and ambivalence towards alternative meat in Korea has been determined.

The results of this study provide useful information to marketers related to alternative meat and conventional meat. Our findings suggest that alternative meat marketers should not address negative phrases like "it is made in a laboratory or by biotechnology" because the variables of unnaturalness and distrust of biotechnology have strong significance in affecting consumers' buying intention. On the other hand, conventional meat marketers should emphasize the naturalness of conventional meat for the same reason. Meat consumption is based on various culture, so the author of this study believes that the present results help an integral understanding of alternative meat, focusing on consumers' psychological states.

**Author Contributions:** Conceptualization, J.H., J.M. and J.J.; methodology, J.H., J.M. and J.Y.; software, J.H. and J.Y.; data collection, J.H., J.Y. and J.M.; writing—original draft preparation, J.H.; writing—review and editing, J.M., J.Y. and J.J. All authors have read and agreed to the published version of the manuscript.

**Funding:** This research received no external funding.

**Conflicts of Interest:** The authors declare no conflict of interest.

## Appendix A  Survey Items

**Table A1.** Construct.

| Construct | Indicator |
|---|---|
| Sustainability of animal production system | X1: Factory farms seem to elevate the risk of novel viral outbreaks<br>X2: Antibiotic-resistant bacteria that arise in factory farms can spread to human hosts.<br>X3: If people are concerned about the treatment of animals, or the threat of zoonotic epidemics and antibiotic resistance, they should change their consumption. |
| Unnaturalness | X4: Our lives start with a single cell, which is undoubtedly very natural. Cultured meat/plant-based meat originates from a single cell, just as the plants that we eat.<br>X5: Cultured meat/plant-based meat is more natural than conventional meat<br>X6: Eating human-made meat is an unnatural practice that separates us further from nature. |
| Drug-free cleanness | X7: In recent years, I have tried to limit my red meat consumption because farmers use antibiotics for treating animals.<br>X8: In recent years, I have tried to limit my red meat consumption because hygienic conditions are poor.<br>X9: In recent years, I have tried to limit my red meat consumption because of hormone residues. |
| Distrust of biotechnology | X10: Technologically genetically modified cells can trigger environmental disasters when they go from laboratories to the outside world.<br>X11: Gene technology is forced on us. We do not have any chance to avoid this technology in the future.<br>X12: The fear of gene technology is a psychological problem; people have always been afraid of new technologies |
| Food curiosity | X13: When you prepare or when you eat a food that you know, do you love to add new ingredients?<br>X14: Do you like to know what is in a dish?<br>X15: When you eat at home, do you take the time to look, feel, and touch what you are going to eat? |
| Food neophobia | X16: I do not trust new foods.<br>X17: If I do not know what is in a food, I will not try it.<br>X18: I am afraid to eat things I have never had before. |

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
