# Peer review of "Factors Affecting Consumers’ Alternative Meats Buying Intentions: Plant-Based Meat Alternative and Cultured Meat"

_sustainability, doi:10.3390/su12145662_

Round 1

Reviewer 1 Report

Author has been made a great effort to improve the manuscript, however some corrections and response to comments are required.

  • Methodology – what was the low response rate? (drop out)
  • Lines 220-221 - What was the validity and reproducibility? Please provide the data.
  • First table should provide demographic information (gender, age, place of residence, etc.). (please replace table no 3) .
  • All abbreviations in tables should be explained in legend.
  • Figure 2 is poor resolution
  • Table 2 is difficult to follow due to the some shift in data. This table should be improved
  • The discussion section should be still improved. Some sentences are not communicative enough (e.g. “This result is consistent with Verbeke et al. [37].”/ “In terms of food neophobia, our results are the same as Wilks et al. [20]’s.”). Some detailed information about other studies are necessary. Authors should add the appropriate references in this section. Moreover, authors should emphasize the novelty of the study, due the fact, that in the presented version of the discussion majority of the findings are same as in the study of others authors, so the novelty of the conducted research are needed.
  • Conclusion should be shortened and be more focused on specific findings of the study.

Author Response

We greatly appreciate your comments which have helped to improve our manuscript. Please see the attachment. Thank you.

Reviewer 2 Report

This paper is of general interest to the readers of this journal. However it must be improved in a several ways, with much more attention paid to ensuring accuracy and appropriate use of language.  It is quite sloppy in how arguments are made and information presented.  Its novelty needs to be more explicit.  What is the paper providing, other than comparing two things which were previously deal with separately?

For example in the introduction, it is stated that "Culturing meat makes it possible to generate skeletal muscle and bio-artificial muscles, which are a key source of animal protein.  Not only is this grammatically incorrect, it is factually incorrect.  Such muscles are not "key sources" of animal protein.  They are only "potential sources of human protein" currently.

The conceptual framework section needs to be tightened up, with much more careful attention paid to the use of terminology, e.g. "feelings" and "attitudes are used interchangeably which is incorrect.  One-sided statements are presented without qualification, e.g. the statement "Alternative meats are more sustainable compared to farm-grown meat products considering the environment, animal welfare, and natural resources and they are good for society" is subject to argument by other authors.  It would be more acceptable, as a scientist, to say "Alternative meats are considered...  The statement "As a result of integrating previous studies, the features of alternative meat can be classified into three overall categories: ethical viewpoint, food safety, and initial reaction" required elaboration - which studies, how did you identify these "categories" from these studies?  Clarification/justification is required for the following statement "In this study, the category of ethical viewpoint is divided into the concept of sustainability (positive cognitive) and unnaturalness (negative cognitive)". Cite some literature for example which shows that unnaturalness is identified as morally unacceptable.

In the discussion, the following sentence does not make sense "They addressed the fact that acceptin cultured meat is not for a few personal benefits but for bigger societal benefits, like "Not in My Back Yard" (NIMBY). "  The relevance of NIMBY is not clear.

Additional references which may be helpful are

CONNER, M. & ARMITAGE, C. J. 2011. Attitudinal ambivalence. Attitudes and Attitude Change.

MAIO, G. R., BELL, D. W. & ESSES, V. M. 1996. Ambivalence and persuasion: The processing of messages about immigrant groups. Journal of Experimental Social Psychology, 32, 513-536.

MAIO, G. R., ESSES, V. M. & BELL, D. W. 2000. Examining conflict between components of attitudes: Ambivalence and inconsistency are distinct constructs. Canadian Journal of Behavioural Science, 32, 58-70.

THOMPSON, M., P. ZANNA, M. & GRIFFIN, D. 1995. Let’s not be indifferent about (attitudinal) ambivalence.

The whole paper needs to be carefully edited for English

Author Response

(The authors gave the same response as above.)

Round 2

Reviewer 1 Report

Author has been made a great effort to improve the manuscript, however some corrections and response to comments are required.

  • Table 1 – please add the number of respondents
  • Table 2 – there are some problems with lines  
  • Taking into account the fact, that the questionnaire was not validated (only pilot test was conducted), the proper information in limitation section should be presented in limitation

Author Response

We would like to thank the reviewer for a careful and thorough reading of this manuscript and for the thoughtful comments and constructive suggestions, which help to improve the quality of this manuscript. Please see the attachment.

This manuscript is a resubmission of an earlier submission. The following is a list of the peer review reports and author responses from that submission.

Round 1

Reviewer 1 Report

Please see attached report which indicates improvements required to the introduction and methodology sections.

Factors Affecting Consumers’ Alternative Meat  Buying Intentions: Cultured Meat and Plant-Based  Meat

Reviewers report

This paper is of interest to the journal’s readership as it is concerned with consumer perspectives on emerging/novel protein sources.  The authors have undertaken primary data collection in the form of an online survey and achieved a large sample size (n=1,017).  My concerns primarily relate to the design and implementation of the online survey.  In addition I have concerns about the argumentation provided in the introduction. 

In the introduction, there are several examples of causality which are not justified and/or sweeping statements. The whole introduction section needs to be fully reviewed and revised to ensure greater precision of the points made and to reflect more a more balanced perspective on the part of the authors.  The authors are referred to the Foods paper by Henchion et al, 2017 available at https://www.ncbi.nlm.nih.gov/pubmed/28726744

·         For example, the first sentence says that consumption of red and processed meat has increased because of its nutritional characteristics and favourable taste.  With regards to nutritional aspects, most authors would distinguish between the nutritional aspects of red and processed meats.  Furthermore, the WHO for example would argue that processed meat does not have positive nutritional characteristics.  The Lancet Commission argue that red meat is unhealthy https://www.thelancet.com/commissions/EAT

·         Another example is the statement that alternative meat is “highly anticipated to reduce problems created by conventional meat”.  This is a very strong statement that would be better presented as “proposed as an opportunity to address some of the problems created by conventional meat production and consumption”.

·         The statement that alternative meat (e.g. cultured meat and plant-based meat) could solve consumers’ concerns regarding drugs and met their health needs ignores the significant volumes of agrichemicals used in the production of crops.

Greater clarity is required regarding the methodology.  It is not clear how participants were identified (was an agency that maintains a consumer panel engaged or was the questionnaire publicised far and wide?) and whether an incentive was offered or not.

It is stated that there were two separate questionnaires with approximately 500 respondents for each questionnaire.  As comparisons are being made between the two groups of respondents, the basis for comparison needs to be made.  In this regard, a profile of the respondents in each group is required at least to ensure that they are comparable in terms of age, gender, etc.  Furthermore, the profile of each sample should be compared to the national population to indicate whether or not they are representative of the national population.

The respondents are Korean however it is not clear whether the questionnaire was in English or Korean.  Translation issues need to be addressed in either case.

The measures identified are plausible however how the various scales were adapted needs to be elaborated to ensure they are still valid.  

The language used in the statements is somewhat concerning, e.g. “zoonotic epidemics” – are the authors confident that their lay respondents would understand such terminology?

Statements are not simple statements, i.e. respondents may agree with one part of the statement and disagree with the other part, and hence it is impossible to interpret their responses.  For example, consider the statement “Eating meat is an unnatural practice that separates us further from nature”.  A respondent could agree that eating meat is an unnatural practice but may not agree that it separates us further from nature.  Statements are often compound or complex statements rather than simple statements

Some statements are leading, e.g. “Our lives start with a single cell, which is undoubtedly very natural.  ….

The drug-free cleanness construct is a particular problem as this does not make sense if a respondent has not tried to limit their red meat consumption and/or in fact increased their red meat consumption.  This question only applies to consumers who tried to limit their red meat consumption.  From my reading of Ergonul (2013), who is the cited source of this scale, statements regarding self-reported behaviour have been adapted in a leading way without clear justification.  Table 3 in Ergonul (2013) reports on a question regarding “Farmers use antibiotics for healing the animals”, whereas the statement used in the Foods paper is “Farmers use antibiotics for treating animals”. 

It is claimed that the “sustainability of animal welfare scale” was expanded from Anomaly (2015).  I read this paper and find no evidence of any scale in the paper. It appears that the authors developed statements based on this paper rather than adapted/expanded an existing scale.  The same is the case for the unnaturalness scale which is reported to be adapted from Welin and Van der Weele.

It is not meaningful to comment on the analysis, results and discussion in the paper as the methodology is fundamentally flawed.

Reviewer 2 Report

The manuscript entitled “Factors Affecting Consumers’ Alternative Meat Buying Intentions: Cultured Meat and Plant-Based Meat” presents an interesting issue, however it requires some amendments. 

Title

It should be corrected - Plant-Based Meat (is misleading) – it should be Plant-Based Meat Alternatives/ meat Replacement Plant-Based Food or similar. The same correction should be included to the main body of the study.

Abstract:

- Line 22 – please specify the number of respondents in each survey (n=513 and n=504)

Introduction:

- Authors presented two main alternatives for meat - plant-based and cultured. Moreover authors stated that “cultured meat is not commercialized yet”, simultaneously no information was presented associated with insect-based products, which may be also an alternative protein source. This is important due to the fact, that insect-based products are commercially available, therefore it will be more suitable to compare such products also. It should be at least mentioned t in this section.  

- Line 71-72 – “In terms of the advantages of meat, it tastes good and is nutritionally is nutritionally beneficial for consumer health [24].” This is not always true. Nowadays there are more evidences that red meat and processed meat are more harmful than beneficial for human health (potential adverse health effects of red meat consumption on major chronic diseases, such as diabetes, coronary heart disease, heart failure, stroke and cancer at several sites, and mortality – Wolk A. Potential health hazards of eating red meat. J Intern Med. 2017 Feb;281(2):106-122. doi: 10.1111/joim.12543; enhances the risks of chronic ill health, such as from colorectal cancer and cardiovascular disease – Godfray HC, Aveyard P, Garnett T, Hall JW, Key TJ, Lorimer J, Pierrehumbert RT, Scarborough P, Springmann M, Jebb SA. Meat consumption, health, and the environment. Science. 2018 Jul 20;361(6399). pii: eaam5324. doi: 10.1126/science.aam5324.) Sentence should be corrected. 

- Figure 1 – according to the food neophobia – If the authors compared plant and meat-based products the effects of food neophobia will be for animal-derived product (not only cultured meat but regular one) (please see: Alley, T. R., & Potter, K. A.  (2011).  Food neophobia and sensation seeking.  Pp. 707- 724 in V. R. Preedy, R. R. Watson  & C. R. Martin (eds.), Handbook of Behavior, Food and Nutrition.  Springer.)  This approach should be reconsidered (as a bias).  

- Line 126 – “…which can kill people” This sentence should be rewritten – more scientific language should be used (here and anywhere else)

- Linea 141-144 – These senesces are not well connected – should be rewritten. 

Materials and methods:

- This section is not informative enough – there is almost no information about methodology. 

- More information must be provided about the study recruitment.

- What were the criteria of recruitment? 

- What were the criteria of exclusion and inclusion to the study? 

- How was the link with internet survey provided to the respondents and where? 

- Please add the information about number of ethics commission approval

- How was informed consent of the respondents collected?

- Was the questionnaire previously validated? What were the accuracy and consistency of this questionnaire? This issue is especially crucial to obtain the reliable results.  Please add the appropriate statistical analyses.

- Please add more information about tool – how were they translated and validated? Authors must present the proper translation (conducted by bilingual translator or  by at least two independent translators) and verification as well as validation of this questionnaire for this new translation (authors can see publication: Tsang, S., Royse, C. F., & Terkawi, A. S. (2017). Guidelines for developing, translating, and validating a questionnaire in perioperative and pain medicine. Saudi Journal of Anaesthesia, 11, S80-S89.)

- Table 1 – for food neophobia scale (FNS) – authors presented not original statements from the scale but, some translation (was forward and backward translational applied?). Necessary information must be presented and discussed  – how it could influence the results? For FNS as for other tools this issue must be properly explained, otherwise the obtained results are not reliable.

Results:

- The table with socio-demographic data of respondents should be presented. 

- Line 197 – “To assess the validity of each individual item, this study examined internal consistency measures using the composite reliability.” What did authors mean by “item”? It is unappropriated to assess validity for items, but it should be rather assessed for a scale. It should be verified and corrected. 

- Table 2 – p-Value and the significance should be presented. 

- Table 2 should be corrected, due to the fact, that data is shifted in table cells and it is difficult to follow. 

- Please add the additional statistical analysis associated with socio-demographic co-variables, that could interfere (as e.g. women are more food neophobic than men, women are more disgust sensitive than men, etc.). 

Discussion:

- Line 282-328 – In this part, there is no discussion, but authors rather divagated about the study. This section must be totally rewritten. Authors should relate the findings to those of similar studies and point the differences and similarities between the studies. Authors should add the appropriate references to this section.

Conclusion

- Conclusion should be focused only on the findings of the study (e.g., First part is not associated with the study). 

References:

- Should be corrected according the author guideline.